

# Beyond Gross-Pitaevskii equation for 1D gas: Quasiparticles and solitons

Jakub Kopyciński[1*], Maciej Łebek[1,2], Maciej Marciniak[1],
Rafał Ołdziejewski[3,4], Wojciech Górecki[2] and Krzysztof Pawłowski[1]

**1** Center for Theoretical Physics, Polish Academy of Sciences,
Al. Lotników 32/46, 02-668 Warsaw, Poland
**2** Faculty of Physics, University of Warsaw, Pasteura 5, 02-093 Warsaw, Poland
**3** Max Planck Institute of Quantum Optics, 85748 Garching, Germany
**4** Munich Center for Quantum Science and Technology,
Schellingstrasse 4, 80799 Munich, Germany

⋆ jkopycinski@cft.edu.pl

## Abstract

Describing properties of a strongly interacting quantum many-body system poses a serious challenge both for theory and experiment. In this work, we study excitations of one-dimensional repulsive Bose gas for arbitrary interaction strength using a hydrodynamic approach. We use linearization to study particle (type-I) excitations and numerical minimization to study hole (type-II) excitations. We observe a good agreement between our approach and exact solutions of the Lieb-Liniger model for the particle modes and discrepancies for the hole modes. Therefore, the hydrodynamical equations find to be useful for long-wave structures like phonons and of a limited range of applicability for short-wave ones like narrow solitons. We discuss potential further applications of the method.



# 1  Introduction

In weakly interacting ultracold Bose gas, the mean-field approach given by a single particle non-linear Schrodinger equation, which is also known as the Gross-Pitaevski equation (GPE), has explained and predicted a large swathe of phenomena [1]. Interestingly, the GPE derives from the complete neglecting of the mutual quantum correlations between the particles, and yet it describes non-linear phenomena that originate from interactions between them. The most known are solitons observed in experiments with an ultracold gas confined in a steep cigar-shaped harmonic trap [2–4]. In the repulsive gas, the solitons are density dips travelling with a constant speed, robust due to the balance between interaction and dispersion. The dark solitons predicted by the GPE correspond to hole excitations in the many-body system described by the linear Lieb-Liniger model [5]. The above correspondence has been debated for many years before being fully justified only recently by works of different authors in Refs. [6–14]. Should the correspondence hold for stronger interactions, however, remains an open question.

The original GPE may overlook interesting physics even for weakly interacting systems with quantum depletion [15,16] still being very small. In the presence of both repulsive and attractive interparticle forces of comparable interaction strength, the mean-field contributions to the total energy almost cancel each other out and become of the same order as low-energy quantum fluctuations. Their sudden prominence has given rise to the discovery of quantum droplets and supersolids that have been recently studied at length both in experiment and theory [17]. In three dimensions, one can easily add the effective term describing quantum fluctuations, called the Lee-Huang-Yang (LHY) correction, to the GPE leading to its extended version both for Bose-Bose mixtures [18] and dipolar Bose gas [19,20], that resolves the initial problem. For the dipolar case in one or two dimensions, however, such an approach is more challenging to justify [21]. Nevertheless, in one dimension, both dipolar and Bose-Bose mixtures have been investigated already by using ab-initio approaches like Monte-Carlo or exact diagonalization, for example in Refs. [22–25].

The widely-used GPE, even supplemented by the LHY correction, becomes unreliable for the strong interaction, especially in lower dimensions where quantum effects are enhanced. A question arises whether there exists a better effective non-linear model that would be useful to study quasi-1D Bose gas also for strongly correlated systems. The simplest system to study is the uniform one and interacting only via short-range repulsive forces. The underlying many-body model describing such a system with $N$ particles is given by the exactly solvable Lieb-Liniger (LL) model [26–28]. Nowadays, the LL model is a testbed for many-body methods, a starting point of subsequent theoretical frameworks (like Luttinger liquid) but also an active research area in mathematical physics. [29,30].

Although the LL model describes uniform repulsive 1D Bose gas completely, an effective framework, somehow similar to the GPE, has been sought that would be suitable also for

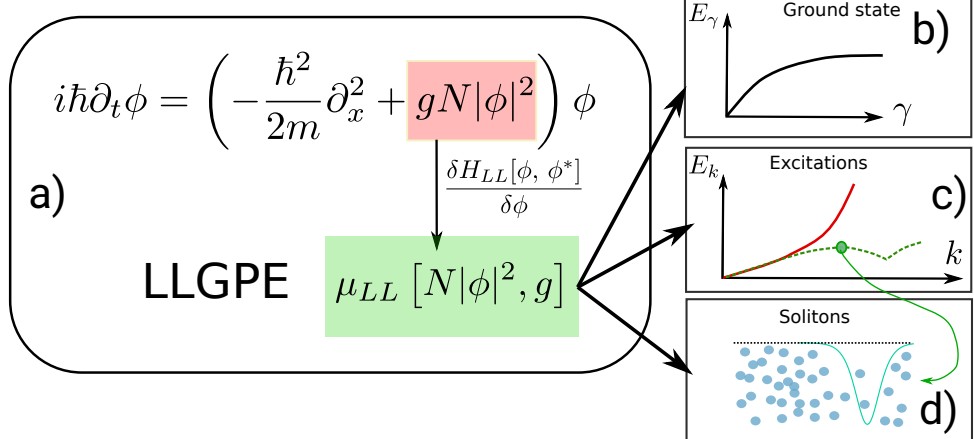

Figure 1: Graphical abstract: a) We describe 1D repulsive Bose gas using an equation called here the LLGPE. The LLGPE differs from the GPE only by the interaction energy term, which, according to the explanation given in this work, is replaced with the chemical potential of the Lieb-Liniger model. b) We study the energy of the ground state. c) We analyse the dispersion relations of both types of the Lieb-Liniger elementary excitations: the type-I excitations (Bogoliubov-like, solid red line) and type-II excitations (solitonic branch, dashed green). d) We study spatial properties of the type-II excitations that correspond to the black solitons and compare them with solitonic solutions of the GPE and LLGPE equations.

In all these tasks (b-d), we benchmark our findings with the GPE and exact solutions of the Lieb-Liniger model for all interaction regimes, from the weakly interacting gas up to the Tonks-Girardeau limit.

strongly correlated systems with additional trapping potential or different interaction type. In the extreme case of trapped 1D gas in the Tonks-Girardeau limit, Kolomeisky et al. [31] already proposed a non-linear equation with higher-order nonlinearity than the GPE. Despite the initial critics by Girardeau and Wright [32], the equation was generalized in [33,34] to any interaction strength and successfully applied to a problem of an expanding 1D cloud. Similar extensions served to study ground state properties as well as collective excitations of strongly interacting gas in a harmonic trap [35, 36] and dynamics of shock waves [37, 38]. Finally, alternatives of the GPE for strongly interacting dipolar systems have been proposed [23, 39] and have already shown usefulness in timely topics like dipolar quantum droplets [23, 40].

To assess the validity scope of the effective approach employed in this work, we focus on the system described by the LL model. We will benchmark the ground state and its elementary excitations inferred from the effective non-linear approach [23, 31, 34], called here the Lieb-Liniger Gross-Pitaevski equation (LLGPE), against the corresponding solutions of the LL model [26, 27] to test the validity range of the former. Our equation has the form of the GPE but with the nonlinearity from the exact results for the Lieb-Liniger model, see Fig. 1. It has appeared in the literature under the names the Modified Non-linear Schrödinger Equation [41] and the Generalized Non-linear Schrödinger Equation [42]. Here, we will focus on two branches of elementary excitations in the LL, i.e. particle (type-I) and hole (type-II) modes that correspond to phonons and solitons respectively. Their hallmarks are their dispersion relations and, for the type-II excitations, their spatial dependence. We will discuss the interpretation of solitons as the many-body excitations for strong and intermediate interaction strength. In this way, we will test the scope of the non-linear model, which is the cornerstone of the quantum droplet description employed in [23] and an analysis of breathing modes in [40].

The paper and the main results are organized as follows. In Sec. 2 we introduce all impor-

tant models: the LL model, the standard non-linear equation and finally the main subject of this paper, i.e. the generalized non-linear equation. We discuss its derivation and the ground state wave-function. In Sec. 3, we focus on particle excitations. Linearizing the generalized non-linear equation, we calculate the excitation spectrum and the sound velocity for all $\gamma$ that coincide with many-body results of the LL model. Subsequently, we compare our results to the standard GPE and examine its validity range. In Sec. 4, we analyse dark solitons predicted by the generalized non-linear model. In the first step, we compare our findings with solitons obtained for two limiting cases of $\gamma \to 0$ and $\gamma \to \infty$ within corresponding effective, non-linear models. Our model reproduces the preceding results and predicts solitons for intermediate values of $\gamma$. Later, we examine the correspondence between hole excitations in the LL model and the dark solitons from the generalized non-linear equation for any $\gamma$. As $\gamma$ increases, narrow solitons predicted by the hydrodynamical approach lose their connection with the hole excitations in the LL model. In Sec. 5, we predict the scope of validity of the generalized equation. The hydrodynamical equations find to be useful for long-wave structures like phonons and of a limited range of applicability for short-wave ones like narrow solitons. In Sec. 6, we summarize the main findings of the work and discuss further applications of the generalized non-linear equation.

## 2 Models

In this Section, we introduce different models of $N$ bosons moving along a circle of length $L$ and interacting via delta potential $V(x) = g \delta(x)$, where $x$ is a distance between particles and $g$ is a coupling strength.

The fundamental model is expressed by the following many-body Hamiltonian:

$$\hat{H} = -\frac{\hbar^2}{2m} \sum_{j=1}^{N} \partial_{x_j}^2 + g \sum_{\substack{j,l \\ j<l}}^{N} \delta(x_j - x_l), \tag{1}$$

where $x_j$ denotes position of the $j$-th particle. The Hamiltonian (1) equals to the Lieb-Liniger Hamiltonian, with physical constants written explicitly.

The seminal papers of Lieb and Liniger [26, 27] present the general form of all eigenstates of the Hamiltonian (1) in the case of repulsive interactions, i.e. $g > 0$. A useful dimensionless parameter is:

$$\gamma := \frac{m}{\hbar^2} \frac{g L}{N}, \tag{2}$$

known as the Lieb parameter. In particular, the energy of the ground state $E_0$ in the thermodynamic limit ($N \to \infty$, $L \to \infty$, $N/L = $ const) is given by:

$$E_0[N, L] = \frac{\hbar^2}{2m} \frac{N^3}{L^2} e_{\mathrm{LL}}(\gamma), \tag{3}$$

that defines the pressure [43]:

$$P_{\mathrm{LL}}[N/L] = -\frac{\partial E_0[N, L]}{\partial L} = \frac{\hbar^2}{2m} \frac{N^3}{L^3} \left( 2 e_{\mathrm{LL}}(\gamma) - \gamma e'_{\mathrm{LL}}(\gamma) \right), \tag{4}$$

and the chemical potential:

$$\mu_{\mathrm{LL}}[N/L] = \frac{\partial E_0[N, L]}{\partial N} = \frac{\hbar^2}{2m} \frac{N^2}{L^2} \left( 3 e_{\mathrm{LL}}(\gamma) - \gamma e'_{\mathrm{LL}}(\gamma) \right), \tag{5}$$

as well. Here, $e_{\mathrm{LL}}(\gamma)$ does not have a known explicit analytical form [1,2]. Although the explicit expression for the function $e_{\mathrm{LL}}(\gamma)$ is missing, its very expansions for small and large $\gamma$ are known. In particular, for small $\gamma$, it reads $e_{\mathrm{LL}}(\gamma) \approx \gamma - 4\gamma^{3/2}/(3\pi)$. The first term of the expansion leads to the mean-field energy, whereas the second negative term can be identified with the LHY correction. Note that, unlike the total energy, the pressure and chemical potential as intensive properties depend only on the ratio $N/L$.

Unfortunately, the many-body eigenstates of the LL model [26] are often impractical to use straightforwardly due to the number of permutations of the $N$-particle bosonic state. One has to resort to approximate models, from which the easiest one comprises the GPE. A typical derivation of GPE is based on an Ansatz in the form of the Hartree product, in which a many-body solution of the Schrödinger equation $\psi(x_1, \ldots, x_N, t)$ is approximated by a product state $\prod_{j=1}^{N} \phi(x_j, t)$. The LL Hamiltonian (1) averaged in such Ansatz gives the mean energy[3]:

$$E_{\mathrm{GPE}}[\phi] = \frac{N}{2} \int dx \left[ \frac{\hbar^2}{m} \left| \frac{d\phi}{dx} \right|^2 + gN|\phi|^4 \right]. \tag{6}$$

The least action principle applied to the energy functional above leads to the Gross-Pitaevskii equation for an optimal orbital $\phi(x, t)$:

$$i\hbar \partial_t \phi(x, t) = \left( -\frac{\hbar^2}{2m} \partial_x^2 + gN|\phi(x, t)|^2 \right) \phi(x, t). \tag{7}$$

Most of the phenomena observed in the first years of research on the Bose-Einstein condensate were sufficiently well described by this equation. Despite its indisputable role, however, the GPE equation is only an approximation. For weak interactions, it does not include effects related to the depletion of condensate by interactions, e.g. quantum depletion (including the LHY correction). For stronger interactions, due to the resulting correlations between atoms, the system state is no longer a product state, and the GPE equation (7) is no more justified. On the other hand, there might exist another model that would be able to capture the essentials of the system even in the strongly interacting regime. Indeed, such models for fermions are being derived routinely in the frame of the density functional theory.

Here, we use the hydrodynamical approach to extend approximate models beyond weak interactions. The classical hydrodynamic equations[4] read:

$$\frac{\partial}{\partial t} \rho + \frac{\partial}{\partial x}(\rho v) = 0, \tag{8}$$

$$\frac{\partial}{\partial t} v + v \frac{\partial}{\partial x} v = \frac{1}{m\rho} \frac{\partial}{\partial x} P, \tag{9}$$

where $m \cdot \rho(x, t)$, with $\int \rho(x, t) = N$, is the gas density, $v(x, t)$ denotes the velocity field, and $P(x, t)$ indicates pressure. We treat an atomic cloud as composed of small 'volume' elements, which are still large enough that comprise many particles. Following Ref. [33], we assume local equilibrium, i.e. in each 'volume' element the energy density, pressure and chemical potential are fixed by the corresponding values of the many-body energy of the ground state (3) (after replacing $N/L$ by $\rho(x, t)$ everywhere).

---

[1]The values of $e_{\mathrm{LL}}(\gamma)$ can be determined by solving the Fredholm equations given in [26] (where it is denoted as $e(\gamma)$ without subscripts).

[2]Please do not confuse with $e_{\mathrm{LL}}$ introduced in Ref. [23], where it had a meaning of the total energy per volume unit. Moreover, it used a crude approximation of the total energy.

[3]We consider $N \gg 1$ therefore we replace $N-1$ with $N$.

[4]The quantum hydrodynamic equation might consist of the quantum pressure term. Its role and flows in the considered system are discussed in [37].

Comparing (4) and (5), one may check by direct calculation that the Euler equation (9) may be rewritten as:

$$\frac{\partial}{\partial t}v + v\frac{\partial}{\partial x}v = -\frac{1}{m}\frac{\partial}{\partial x}\left(\mu_{\text{LL}}[\rho]\right).$$ (10)

Note that Eq. (10) may be derived in an alternative, more intuitive way. Suppose the fluid consists of many particles and consider the trajectory $(x(t), t)$ of one of them. As the potential energy for the particle at any point is given by a chemical potential $\mu_{\text{LL}}$, one has directly from Newton's law of motion:

$$\frac{\mathrm{d}}{\mathrm{d}t}mv(x(t), t) = -\frac{\partial}{\partial x}\mu_{\text{LL}}[\rho(x, t)],$$ (11)

which, after a straightforward applying of $\frac{\mathrm{d}}{\mathrm{d}t}$, gives Eq. (10).

Introducing

$$\phi(x) = \sqrt{\rho/N}e^{i\varphi},$$ (12)

with $\hbar\partial_x\varphi = mv$, the two hydrodynamical equations (8),(10) can be written, up to a quantum pressure term, in a compact form [37, 44]:

$$i\hbar\frac{\partial}{\partial t}\phi = -\frac{\hbar^2}{2m}\frac{\partial^2\phi}{\partial x^2} + \mu_{\text{LL}}\left[N|\phi|^2\right]\phi,$$ (13)

with

$$\mu_{\text{LL}}\left[N|\phi|^2\right] = \frac{\hbar^2}{2m}N^2|\phi|^4\left(3e_{\text{LL}}\left(\frac{\kappa}{N|\phi|^2}\right) - \frac{\kappa}{N|\phi|^2}e'_{\text{LL}}\left(\frac{\kappa}{N|\phi|^2}\right)\right),$$ (14)

where $\kappa := \frac{gm}{\hbar^2}$ denotes a parameter of inverse length dimension.

The equation (13) is our main subject of interest. In particular, one can rewrite it also as

$$i\hbar\frac{\partial}{\partial t}\phi = \frac{\delta\mathcal{H}_{\text{LL}}[\phi, \phi^*]}{\delta\phi^*},$$ (15)

where

$$\mathcal{H}_{\text{LL}}[\phi, \phi^*] := \frac{\hbar^2}{2m}\left|\frac{d\phi}{dx}\right|^2 + \frac{\hbar^2}{2m}N^2|\phi|^6 e_{\text{LL}}\left(\frac{\kappa}{N|\phi|^2}\right),$$ (16)

is the energy density. Thus the LLGPE can be derived using the least action principle for the energy functional:

$$E[\phi] = N\int dx\,\mathcal{H}[\phi, \phi^*] = \frac{N\hbar^2}{2m}\int dx\left[\left|\frac{d\phi}{dx}\right|^2 + N^2|\phi|^6 e_{\text{LL}}\left(\frac{\kappa}{N|\phi|^2}\right)\right].$$ (17)

Notably, $\phi$ does not have interpretation of a macroscopically occupied orbital as in the derivation of GPE.

For $\gamma \ll 1$, the function $e_{\text{LL}}(\gamma)$ may be approximated in the first order as $e_{\text{LL}}(\gamma) \approx \gamma$, and then the resulting dynamical equation (13) coincides with the GPE for $\gamma \ll 1$. In this regime, the pressure (4) $P_{\text{LL}}[\rho] = \frac{g}{2}\rho^2$ appearing in the hydrodynamical equation corresponds to the pressure of classical interacting gas.

For the opposite limit of $\gamma \to \infty$, fermionization occurs in one-dimensional Bose gas – the value of $e_{\text{LL}}(\gamma)$ converges to a constant $\frac{\pi^2}{3}$, so (13) coincides with the equation proposed in [31]:

$$i\hbar\frac{\partial}{\partial t}\phi = -\frac{\hbar^2}{2m}\frac{\partial^2\phi}{\partial x^2} + \frac{\hbar^2\pi^2}{2m}N^2|\phi|^4\phi.$$ (18)

In such a case, the pressure (4) $P_{\text{LL}}[\rho] = \frac{4\pi^2}{3}\frac{\hbar^2}{2m}\rho^3$ should be understood rather like an analogy of fermion degeneracy pressure.

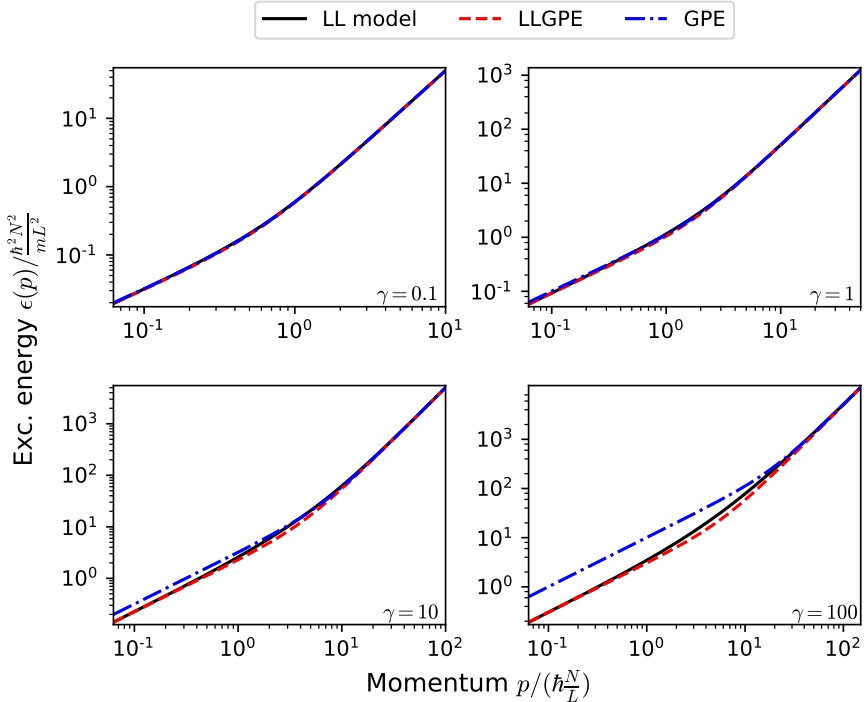

Figure 2: Energy of type-I excitations as a function of the momentum for different values of the interparticle interaction $\gamma$. Following Lieb's recipe [27], excitation energies for the LL model were obtained directly from solutions of the Bethe equations for $N = 100$ particles. On the other hand, red and blue dashed lines correspond to the excitation spectrum calculated from linearization of the GPE (7) and LLGPE, respectively.

From the construction, the minimal value of the energy function (17), obtained for the constant function

$$\phi_{\text{GS}}(x) = \frac{1}{\sqrt{L}}, \tag{19}$$

equals to the actual ground state energy $E_0[N, L]$ for any interaction strength $\gamma$. The ground state of the GPE equals to the function (19) also, but its GPE energy, $E_{\text{GPE}}[\phi_{\text{GS}}] = \frac{1}{2}N^2 g/L$, approximates $E_0$ well in the weakly interacting regime only.

In the following sections, we will benchmark the underlying Lieb-Liniger model with the approximated ones, the GPE and the LLGPE, elaborating the advantages of the latter. In our numerical analysis, we approximate the function $e_{\text{LL}}$, which does not have a known exact and compact form, with a very accurate approximation presented in [45] (compare with Ref. [46]) and repeated here in the Appendix A.

## 3   Phonons and quasiparticles

Sound propagation, superfluidity, normal modes, stiffness — all these depend on the quasi-particles properties in a many-body system. In the case of weak interaction strength, the dispersion relation of excitations was computed by Bogoliubov [47], before the general solution of Lieb. The very notion of quasiparticles, natural in the approximated treatment of Bogoliubov, required a new formulation in the exact many-body theory. Surprisingly, the elementary excitations defined by E. Lieb form two excitation branches instead of one. One of them, the

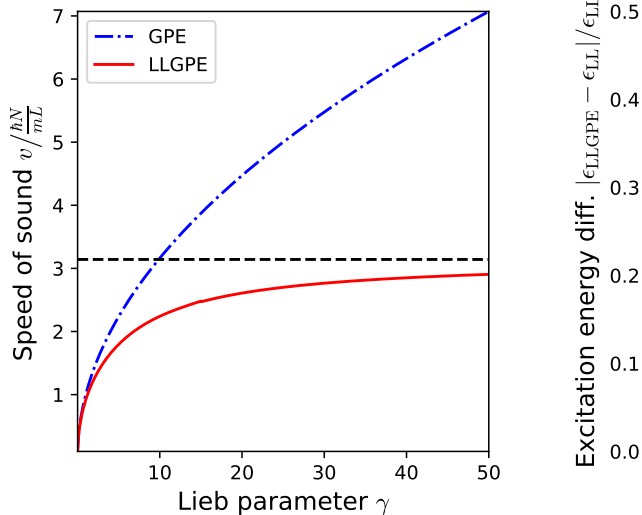 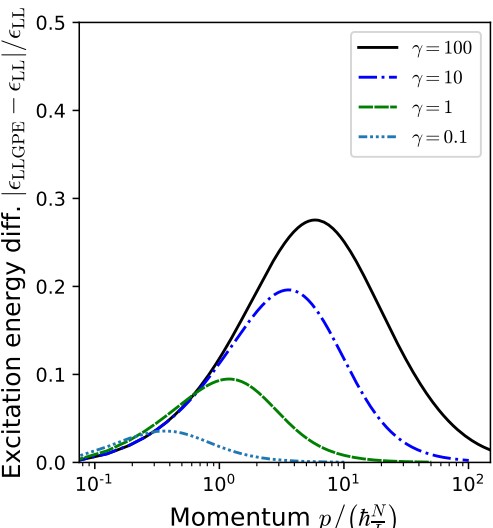

Figure 3: Left: The speed of sound as a function of the Lieb interaction parameter $\gamma$ for two different approaches: LLGPE and GPE. In the former case, the speed of sound has an asymptote $\lim_{\gamma \to \infty} v_{\text{LLGPE}} = \frac{\pi \hbar N}{mL}$, which is marked with the black dashed line. Right: Relative excitation energy difference $\frac{|\epsilon_{\text{LLGPE}} - \epsilon_{\text{LL}}|}{\epsilon_{\text{LL}}}$ as a function of momentum for different Lieb interaction parameters $\gamma$.

branch of the so-called type-I excitations, has almost the same dispersion relation $E_I(p)$ as the Bogoliubov modes (in the weak interaction regime). For low momenta, their energy scales linearly with momentum,

$$E_I(p) \overset{p \to 0}{\approx} v |p|, \tag{20}$$

where $v$ stands for the speed of sound. These excitations, responsible for superfluidity, are the carriers of sound and are known as phonons.

The fast quasiparticles have the dispersion relation of free particles:

$$E_I(p) \overset{p \to \infty}{\approx} p^2/(2m). \tag{21}$$

As shown in Ref. [48], linearization around the ground state in the frame of the GPE leads to the same dispersion relation as using Bogoliubov quasiparticles. Therefore, we recall and apply this method to both approximate models, the GPE and the LLGPE, to trace the differences between, similarly to Ref. [35].

**Linearization.** — We consider a small perturbation to the stationary solution (19). Following Ref. [49], we restrict ourselves to linearized dynamics and propose an Ansatz for a time-dependent solution in the form:

$$\phi(x, t) = \left( \frac{1}{\sqrt{L}} + \delta\phi(x, t) \right) e^{-i\mu_{\text{LL}}[N/L]t/\hbar}, \tag{22}$$

where $\delta\phi(x, t) = \sum_p u_p(x)e^{-i\epsilon_p t/\hbar} + v_p^*(x)^{i\epsilon_p t/\hbar}$ is assumed to be a small correction to the stationary solution (19). Hence, after substituting the Ansatz to the LLGPE, we keep terms at most linear in $\delta\phi$ and obtain

$$i\hbar\partial_t \delta\phi = \left[ -\frac{\hbar^2}{2m}\partial_x^2 + mv_{\text{LL}}^2[N/L] \right]\delta\phi + mv_{\text{LL}}^2[N/L]\delta\phi^*, \tag{23}$$

where $v_{\mathrm{LL}}[N/L] = \frac{\hbar N}{mL}\sqrt{3e_{\mathrm{LL}}(\gamma) - 2\gamma e'_{\mathrm{LL}}(\gamma) + \frac{1}{2}\gamma^2 e''_{\mathrm{LL}}(\gamma)}$ determines an exact expression for the speed of sound in LL model [27, 50]. We use our ansatz for $\delta\phi(x,t)$ getting:

$$\epsilon_p u_p(x) = \left(-\frac{\hbar^2}{2m}\partial_x^2 + m v_{\mathrm{LL}}^2[N/L]\right)u_p(x) + m v_{\mathrm{LL}}^2[N/L]v_p(x), \qquad (24)$$

$$-\epsilon_p v_p(x) = \left(-\frac{\hbar^2}{2m}\partial_x^2 + m v_{\mathrm{LL}}^2[N/L]\right)v_p(x) + m v_{\mathrm{LL}}^2[N/L]u_p(x). \qquad (25)$$

Owing to the translational invariance of our system, we consider $u_p(x) = u_p e^{ipx/\hbar}$ and $v_p(x) = v_p e^{ipx/\hbar}$ simplifying our equations to the form:

$$\epsilon_p u_p = \frac{p^2}{2m}u_p + (u_p + v_p)m v_{\mathrm{LL}}^2[N/L], \qquad (26)$$

$$-\epsilon_p v_p = \frac{p^2}{2m}v_p + (u_p + v_p)m v_{\mathrm{LL}}^2[N/L]. \qquad (27)$$

We solve the above equations, finally obtaining the excitation spectrum:

$$\epsilon(p) = \sqrt{(v_{\mathrm{LL}}[N/L]p)^2 + \left(\frac{p^2}{2m}\right)^2}. \qquad (28)$$

The formula (28) is the main result of this section. In the low momenta limit, one gets $\epsilon(p) \overset{p\to 0}{\approx} v_{\mathrm{LL}}[N/L]|p|$, i.e. the phononic relation with the correct speed of sound; the same as in the many-body approach. Interestingly, we recover also the correct limit (up to the second order) for large momenta (cf. (21)). Thus the linearization of the LLGPE gives a dispersion relation that coincides with the dispersion relation of type-I elementary excitations at low and high energy limit and for any interaction strength. In Fig. 2, we show a comparison between (28) and the exact many-body solutions. The deviations are present only for the intermediate momenta. For completeness, we also show the dispersion relation based on linearization of the GPE. It reads $\epsilon_{\mathrm{GP}}(p) = \sqrt{(v_{\mathrm{GP}}[N/L]p)^2 + \left(\frac{p^2}{2m}\right)^2}$, with a modification in the speed of sound which equals $v_{\mathrm{GP}}[N/L] := \sqrt{gN/mL}$ and tends to inifinity in the Tonks-Girardeau limit. This is illustrated in the left panel of Fig. 3. In the right panel we show relative error between the LLGPE excitations and the type-I excitations. The discrepancies are visible mostly for the intermediate momenta.

The results presented in this section show how well the LLGPE works concerning type-I excitations for all interaction regimes. This leads us to more subtle questions about non-linear effects. Such effects, observed in a number of experiments, appear naturally in the GPE, which is based on the linear equation but are more difficult to identify in the frame of the many-body approach. In the next sections, we shall focus on solitons and their relations with the type-II elementary excitations.

# 4  Solitons

The GPE appears to be very convenient to study non-linear phenomena like solitons. Generally speaking, a soliton is a solution of a non-linear equation in the form of a moving wave, $\phi_S(x,t) = \phi_S(x - v_s t)$, that preserves its shape in dynamics due to the balance between kinetic energy and nonlinearity. In the case of the GPE with repulsive interaction, a soliton has a form of a density dip, as shown by a blue dashed line in Fig. 4. Such solitonic dips, predicted by the

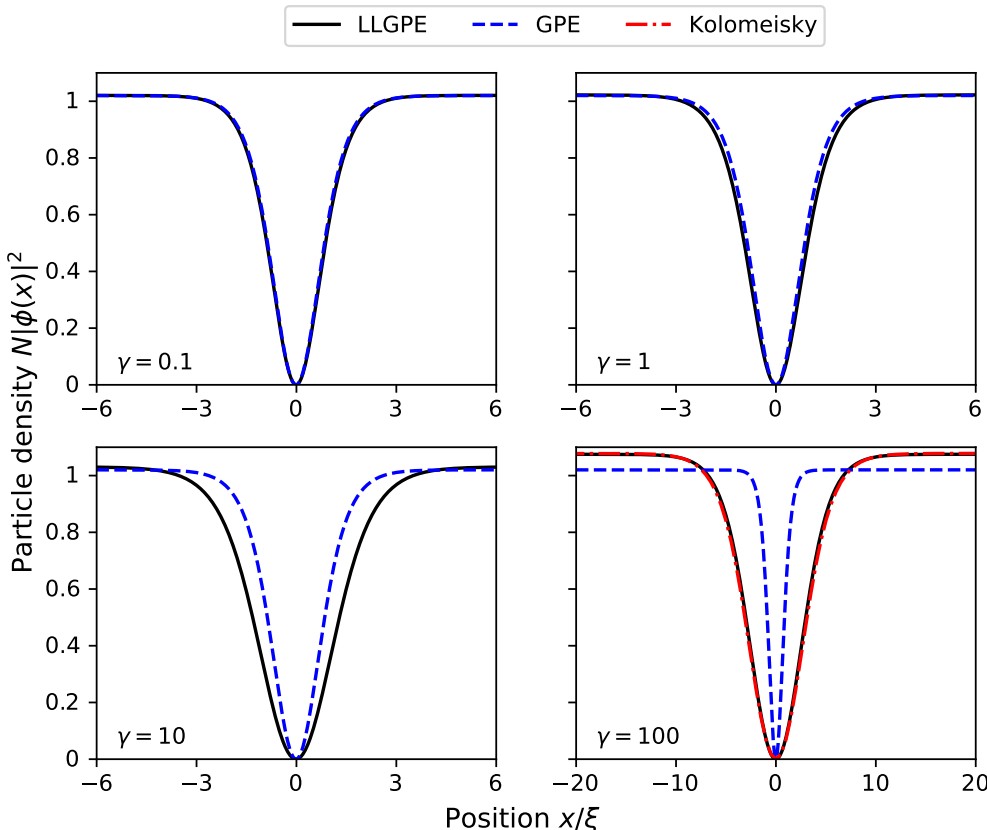

Figure 4: Black soliton density profiles for different equations and interaction strengths. Note the GPE and the LLGPE solutions for $\gamma = 0.1$ as well as LLGPE and Kolomeisky (cf. Ref. [31]) solutions for $\gamma = 100$ overlap each other. In all cases the box size $L = 100\xi$, where $\xi$ is the healing length (which corresponds to $N = 100\sqrt{1/\gamma}$).

GPE, were demonstrated experimentally [2, 3]. Here, we shall focus on a special case, namely solitons with the density dip touching 0, called the *black solitons*.

Besides, the black soliton has a $\pi$ phase jump in the very point of the density minimum. In an infinite box, the black soliton is motionless $v_s = 0$ (the situation in the box with periodic boundary conditions is described in Appendix C). These features are the same as in the solitons in the weakly interacting regime [51].

In this chapter, we discuss i.a. the differences in shape (focusing on the soliton width) between solitons within GPE and LLGPE.

## 4.1 Comparison between solitons of GPE and LLGPE

As discussed above, the linearization of the LLGPE leads to the correct speed of sound in the 1D gas of bosons with short-range interaction, even for the intermediate and strong interaction regime. A natural question arises whether the LLGPE has a solitonic solution beyond the weakly interacting regime.

Here, following the reasoning presented in Ref. [14], we find candidates for solitons in LLGPE numerically for any interaction strength, as the minimal energy states constraint to a $\pi$ jump of phase at the origin. We tested numerically that these states do not move and do preserve their shape in dynamics, as the solitons should do (cf. Figs. 8 and 9 in Appendix C).

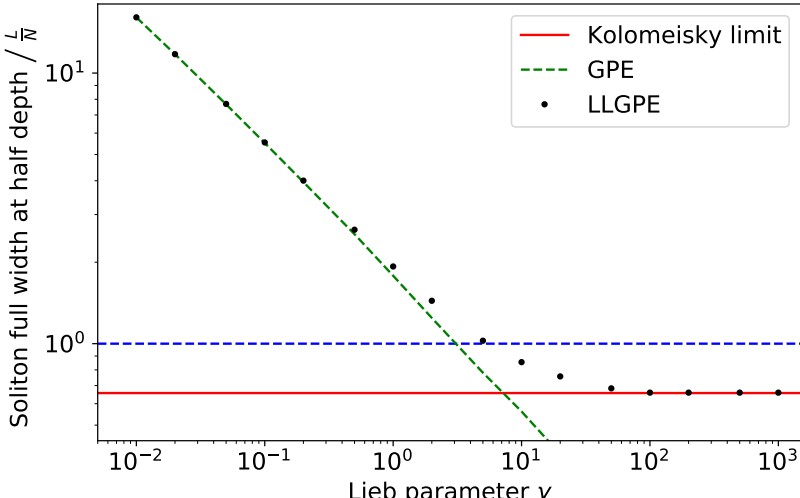

Figure 5: Widths of black LLGPE (black dots) and GPE (green dashed line) solitons found with the ITE method as a function of the Lieb parameter $\gamma$. The dashed blue line corresponds to the average interparticle distance $L/N$ and the solid red line to the Kolomeisky limit for $\gamma \to \infty$ (cf. [31]).

That is the most fundamental property of solitons. The technical details of our method and references to our codes are given in Appendices B and C. The results for the LLGPE are shown in Fig. 4 with the solid black lines, together with the GPE solitons (blue dashed lines). The width of a GPE soliton, which is of the order of the healing length $\xi := \hbar/\sqrt{mNg/L}$, vanishes in the limit of large $g$. This is an unphysical effect: in reality, the gas reaches the Tonks-Girardeau limit, at which the further increase of interaction strength does not affect the system anymore. In the LLGPE, this saturation effect is accounted for by the relation $e_{\mathrm{LL}}(\gamma) \overset{\gamma \to \infty}{\to} \pi^2/3$, and the width of LLGPE solitons converges to a constant. In Fig. 5 we present the widths in the logarithmic scale. As shown in Fig. 4, already for $\gamma = 100$, the LLGPE soliton is indistinguishable from the solitons that are analytically derived in the Tonks-Girardeau limit $\gamma \to \infty$ (red dashed line) [31].

## 4.2 Comparison between solitons and the type-II excitations

It was observed in Ref. [5] that for weak interactions, the GPE solitons have the same dispersion relation as the many-body eigenstates forming the second branch of elementary excitations in the solution of Lieb called in the literature yrast states or type-II excitations or the lowest energy states at fixed momentum. This coincidence was rather unexpected: why the dispersion relation of the solutions of a dynamical, non-linear GPE should match the dispersion of some static solutions of the linear many-body model, distinguished by E. Lieb as the type-II elementary excitations?

There has been a lot of effort devoted to understanding better this relation [6–14, 52, 52–55]. It has been pointed out that the GPE soliton emerges in the high order correlation function computed for the type-II excitation [9, 10, 55, 56]. The other observation points that the solitonic shape appears also in the single-body reduced density matrix evaluated in the appropriate superpositions of the type-II excitations [8, 11–13, 54]. These two different viewpoints were recently unified in Ref. [14]. The agreement holds, however, for weak interactions only, where the GPE is reliable. Here, we address a question whether the relation between solitons and the type-II excitations remains valid for stronger interaction, provided that we use the

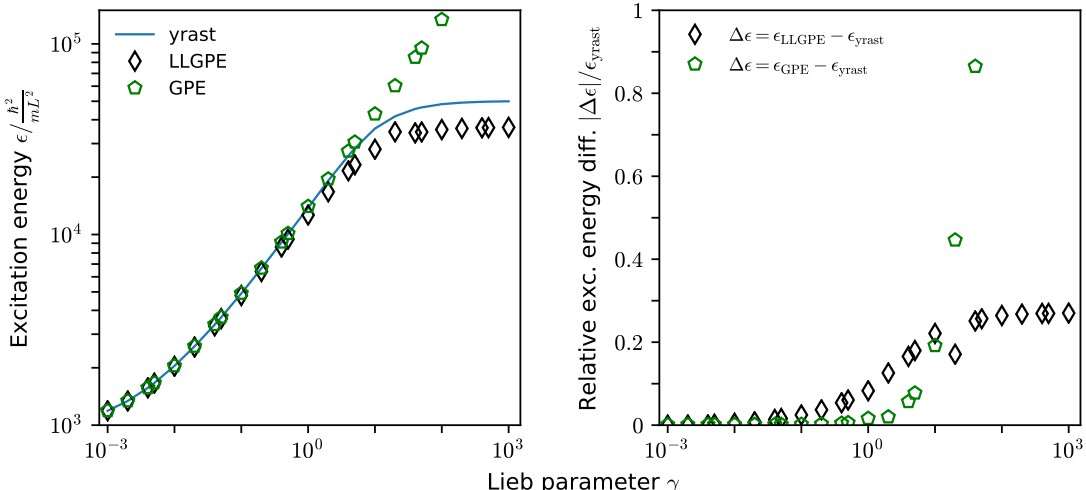

Figure 6: Left: Yrast state excitation energy in the Lieb-Liniger model (blue line) vs dark soliton excitation energy obtained with the LLGPE (black diamonds) and the GPE (green pentagons) as a function of the Lieb parameter $\gamma$. Right: Relative excitation energy $\Delta\epsilon$ of dark solitons within the LLGPE (black diamonds) and GPE (green pentagons). Parameters used in all simulations: $N = 100$.

LLGPE solitons for comparison.

In Fig. 6, we compare the energy of the black GPE and LLGPE solitons with the energies of the type-II excitations evaluated from the exact solution [27] for $N = 100$ particles. For comparison, we use the type-II excitations with the total momentum $\hbar\pi N/L$ that has the same momentum per atom as a black soliton. For a fair comparison, we present the excitation energy being the total energy of the GPE (6) or the LLGPE (17) soliton reduced by the energy of the ground state[5]. The excitation energy of GPE solitons, marked with green pentagons, tends to the infinity with increasing $\gamma$. This is a residue of the vanishing width of the GPE soliton that gives a simple estimation of their kinetic energy by $\frac{\hbar^2}{mw^2}$, where $w$ is the soliton width. In the strong interaction limit, the excitation energy of the LLGPE solitons converges to a constant because the system enters the Tonks-Girardeau phase. The latter dispersion relation qualitatively agrees with the dispersion relation of the type-II excitation, although differences are clearly visible. We show the discrepancies between the energies in the right panel of Fig. 6. The figure shows that in the considered example ($N = 100$), there are significant differences in the relative energies of LLGPE solitons and the type-II excitations, which reach up to 25% in the Tonks-Girardeau limit. Importantly, this divergence is not an artefact of the small number of atoms used in the comparison. In fact, in the thermodynamic limit and for $\gamma \to \infty$, the excitation energy of the yrast state equals $E_{\text{yrast}} - E_{GS} = \frac{\pi^2\hbar^2N^2}{2mL^2}$, whereas the LLGPE soliton has the energy equal to $\frac{\pi^2\hbar^2N^2}{2mL^2}\left[\sqrt{3}\log\left(2+\sqrt{3}\right)/\pi\right]$ [31]. That gives relative difference between energies around 28%.

It is not obvious how to reveal the GPE or LLGPE solitonic density profiles with characteristic dip directly from the type-II "solitonic" excitations. Due to the translational symmetry, the single body density

$$\rho(x) = \int \mathrm{d}x_2 \int \mathrm{d}x_2 \ldots \int \mathrm{d}x_N \, |\psi_E(x, x_2, x_3 \ldots x_N)|^2 \,, \tag{29}$$

of any eigenstate $\psi_E$ of the LL model, including the type-II elementary excitations, is uniform.

---

[5] The subtracted ground state energy differs between the GPE and the LLGPE.

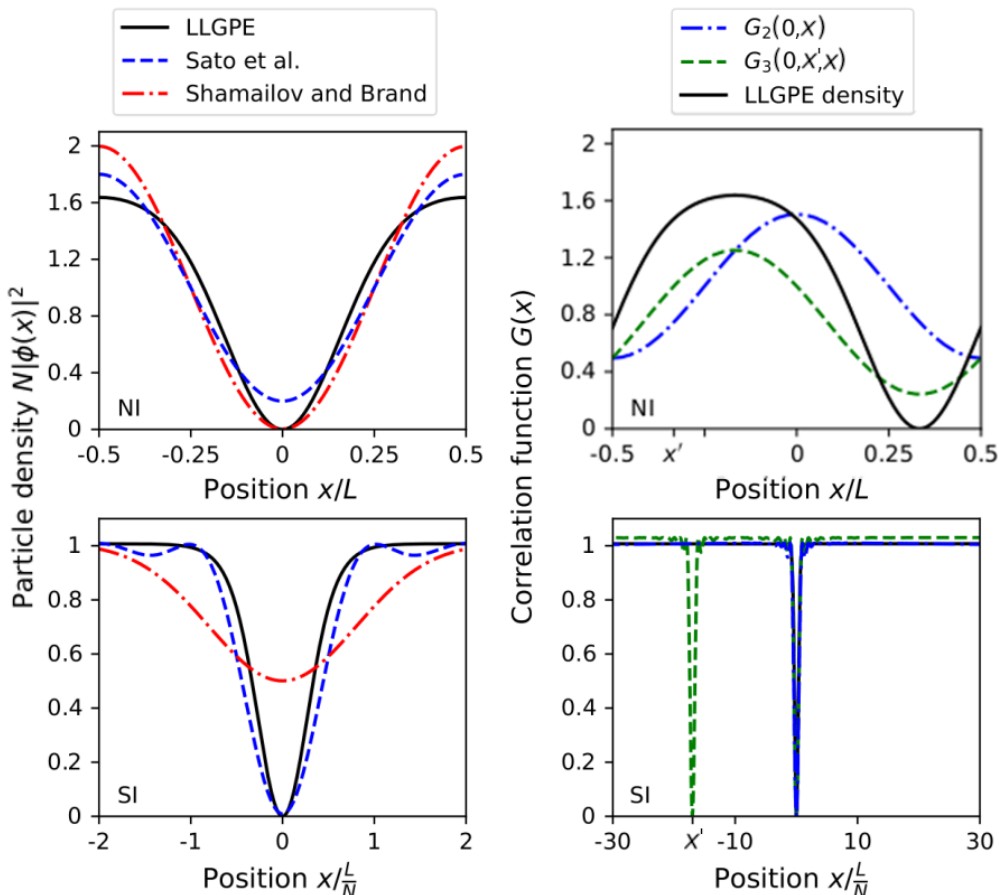

Figure 7: Left panels: Single particle density of the yrast state superpositions (Sato et al. - based on [54], Shamailov and Brand - based on [11] with the dispersion $\Delta P = 0.2\hbar\pi N/L$ used to generate a Gaussian superposition). Right panels: $G_2(0, x)$ and $G_3(0, x', x)$ correlation functions evaluated for the yrast state with the total momentum $\hbar\pi N/L$. NI - non-interacting gas, SI - strongly interacting gas ($\gamma \to \infty$). The $G_2$ function, one dip in $G_3$ and the LLGPE solution overlap each other in the strong interaction limit. The length scales used on the horizontal axes reflect that the soliton width depends only on $L$ in the non-interacting regime, and only on $L/N$ in the strongly interacting regime. The value of $x'$ is chosen randomly.

Therefore, the differences in spatial properties of eigenstates are visible only in typical relations between atoms' positions. Following [9] we study such relations using the families of correlation functions

$$G_m(x_m | x_1, \ldots, x_{m-1}) := \mathcal{N} \left\langle \prod_{i=1}^{m} \hat{\Psi}^\dagger(x_i) \prod_{i=1}^{m} \hat{\Psi}(x_i) \right\rangle, \tag{30}$$

evaluated in the type II eigenstates. The correlation function (30) is a probability density of measuring $m$-th particle at a position $x_m$ provided that $m-1$ particles has been already detected at the random points $x_1, \ldots, x_{m-1}$, and $\mathcal{N}$ stands for a normalization factor. It has been shown in [9,14] that for weak interactions and large $m$, the functions $G_m(x | x_1, \ldots, x_{m-1})$ have spatial profiles close to densities of the GPE solitons. This coincidence is observed for typical positions of other particles $x_1, x_2, \ldots, x_{m-1}$, namely positions drawn according to $G_1$, $G_2, \ldots, G_{m-1}$ respectively.

The other way to reveal mean-field solitons out of the type II excitations was suggested in [8, 11–13, 54]. The Authors studied a wavepacket of the type-II excitations instead of a single one. In this approach, the GPE solitons appear in the single-body density matrix (29). In this approach, different wave packets were proposed.

Here we would like to use these ways, but in the regime of strong-interactions, and compare the results with LLGPE solitons instead of the GPE ones.

In Fig. 7, we recapitulate findings of these two approaches at two extreme regimes: in the limit of non-interacting gas (NI) and the strong-interaction regime (SI). In all panels, we also present LLGPE black solitons marked by a solid black line. The left panels of Fig. 7 show a comparison between the density of the LLGPE soliton and the single-body densities of superpositions of the type-II excitations, as studied in Refs. [11, 12, 54]. These results are not unique in the sense that the reduced density depends on the choice of the superposition.

The right panels of Fig. 7 present benchmarks between LLGPE solitons and $G_m(x_m|x_1, \ldots, x_{m-1})$. Here, we show results for $m = 2$ and $m = 3$ only. A situation in the TG limit (bottom right of Fig. 7) is very different. In the limit $\gamma \to \infty$, the correlation functions assume a simple structure. Owing to the fermionization, the atoms have to be at different places. Thus the probability of finding the $m$-th particle, provided that $m-1$ particles were already measured, has zeros at the locations of already measured particles. Therefore, although the LLGPE soliton seems similar to the $G_2$ function of the yrast state (compare the solid black and blue dashed line in Fig. 7), it has to differ from the higher order correlation functions that have $m$ local minima and not a single one likewise the soliton. Actually, the fact that the LLGPE soliton is close to $G_2$ of the fermionized gas is striking in the context of the hydrodynamical origin of the LLGPE.

# 5 Validity range of LLGPE and solitons

The LLGPE equation derives from the hydrodynamical description, assuming that locally the gas is at equilibrium. However, what *locally* means has to be specified. In the theory of continuous media, one introduces fluid elements consisting of many atoms but spanned over the lengths much shorter than the length scale associated with the density changes. It means that we should restrict our analysis to such solitons for which the latter length, of the order of the soliton width $w$, is much larger than the typical distance between particles, here roughly approximated by $L/N$. The width $w$ can be estimated from the condition that the kinetic energy of a soliton, which is of the order of $\hbar^2/(2mw^2)$, balances the non-linear term in Eq. (13), which is of the order of $\mu_{\mathrm{LL}}[N/L]$. This balance leads to an estimate $w \approx \hbar/\sqrt{2m\mu_{\mathrm{LL}}[N/L]}$. In the case of the weakly interacting limit, i.e. $\mu_{\mathrm{LL}} \approx gN/L$, one gets $w_{\mathrm{GPE}} \approx \hbar/\sqrt{2mgN/L}$, which gives the correct length scale of the GPE soliton width. The hydrodynamical analysis relies on the assumption $w \gg L/N$, which can be expressed as $1 \gg 2\gamma$. Indeed, these qualitative considerations are confirmed by the numerical analysis of the solitonic width illustrated in Fig. 5. The width of the GPE soliton, shown with the dashed green line crosses the typical distance between atoms $L/N$ (blue line) around $\gamma \approx 1$. Apparently, the situation does not significantly change for LLGPE solitons (black dots).

The qualitative discussion of the length scales leads to the conclusion that the solitons presented in the previous section fulfil the assumptions underlying the LLGPE only for weakly interacting gas. In particular, the analytical solutions given in Ref. [31] can go beyond the validity range of the LLGPE. In the Tonks-Girardeau limit considered in Ref. [31], the solitonic width equals the typical distance between particles that, owing to the fermionization, is of the order of $L/N$. In that situation, one cannot define elements of fluid consisting of many atoms with a smooth density profile as assumed in the derivation of the LLGPE. This will be also the

case of other states involving length scales shorter than interparticle distance, for example, two interfering atomic clouds [32] or shock waves [37]. The question of whether solitons exist in the intermediate and strongly interacting regimes may be difficult to resolve on theoretical grounds. We leave the question about their existence open as a challenge for experimentalists.

## 6 Conclusions

The purpose of this work has been to benchmark the generalization of GPE, called here the LLGPE on the well-studied case of gas with solely contact interactions trapped in a 1D box with the periodic boundary conditions. Our analysis support the credibility of this equation, giving the stronger foundations for the further extension, like the one presented in Ref. [23] about quantum droplets in 1D.

In introducing the LLGPE, we invoke the hydrodynamic interpretation. We expect that the equation works in cases where the gas density is slowly varying in space. Indeed, the linearization of the LLGPE leads to the analytical formula for phononic spectra that coincides with the exact formulas known for type-I excitations of the Lieb-Liniger model, even in the limit of infinite interactions. We also find solitonic solutions of the LLGPE and compare their dispersion relation and the spatial dependence to type-II excitations of the Lieb-Liniger model forming a 'solitonic' branch. The correspondence between LLGPE solitons and type-II excitations is limited to weak interaction $\gamma \lesssim 1$. For stronger interaction, the LLGPE solitons do not meet assumptions underlying the LLGPE. The question of whether solitons exist in the 1D gas beyond the weakly interacting regime remains open. The problem might be addressed by experimenters with the technique that was successfully employed for the weakly interacting gas.

To conclude, the LLGPE offers an alternative tool to the GPE, useful in a wide class of smooth solutions even in the strongly interacting 1D Bose gas. In principle, one could also apply it to a gas confined by a slowly varying external potential or for atoms interacting via smooth non-local potential.

## Acknowledgements

We thank G. Astrakharchik and B. Julia-Diaz for fruitful discussions. We wish to thank Z. Ristivojevic for pointing Refs. [46] and [57] to us. Center for Theoretical Physics of the Polish Academy of Sciences is a member of the National Laboratory of Atomic, Molecular and Optical Physics (KL FAMO).

**Funding information** J.K., M.Ł., M.M., K.P. acknowledges support from the (Polish) National Science Center Grant No. 2019/34/E/ST2/00289. This research was supported in part by PLGrid Infrastructure.

## A  Accurate approximations for $e_{\mathrm{LL}}$

The equation studied in this paper is based on the function $e_{\mathrm{LL}}(\gamma)$, part of the ground state energy in the LL model. There is no simple compact formula for this function, but there are known accurate analytical [46] and numerical [45] approximations. In our numerical tools we have used the approximation of $e_{\mathrm{LL}}$ as given in Ref. [45]. In the regime of weak interactions

$\gamma < 1$ we have

$$e_{\mathrm{LL}}(\gamma) = \gamma - \frac{4}{3\pi}\gamma^{3/2} + \left[\frac{1}{6} - \frac{1}{\pi^2}\right]\gamma^2 - 0.0016\gamma^{5/2} + O(\gamma^3). \tag{31}$$

For intermediate interactions $1 \le \gamma < 15$

$$e_{\mathrm{LL}}(\gamma) \approx \gamma - \frac{4}{3\pi}\gamma^{3/2} + \left[\frac{1}{6} - \frac{1}{\pi^2}\right]\gamma^2 - 0.002005\gamma^{5/2} + 0.000419\gamma^3 - 0.000284\gamma^{7/2} + 0.000031\gamma^4. \tag{32}$$

Finally, nearly the fermionized regime $\gamma \ge 15$

$$
\begin{aligned}
e_{\mathrm{LL}}(\gamma) \approx \frac{\pi^2}{3}\Bigg( & 1 - \frac{4}{\gamma} + \frac{12}{\gamma^2} - \frac{10.9448}{\gamma^3} - \frac{130.552}{\gamma^4} + \frac{804.13}{\gamma^5} - \frac{910.345}{\gamma^6} - \frac{15423.8}{\gamma^7} \\
& + \frac{100559.}{\gamma^8} - \frac{67110.5}{\gamma^9} - \frac{2.64681 \times 10^6}{\gamma^{10}} + \frac{1.55627 \times 10^7}{\gamma^{11}} + \frac{4.69185 \times 10^6}{\gamma^{12}} \\
& - \frac{5.35057 \times 10^8}{\gamma^{13}} + \frac{2.6096 \times 10^9}{\gamma^{14}} + \frac{4.84076 \times 10^9}{\gamma^{15}} - \frac{1.16548 \times 10^{11}}{\gamma^{16}} \\
& + \frac{4.35667 \times 10^{11}}{\gamma^{17}} + \frac{1.93421 \times 10^{12}}{\gamma^{18}} - \frac{2.60894 \times 10^{13}}{\gamma^{19}} + \frac{6.51416 \times 10^{13}}{\gamma^{20}} \\
& + O\left(\frac{1}{\gamma^{21}}\right)\Bigg).
\end{aligned}
\tag{33}
$$

After having prepared our manuscript we got aware of better, analytical expansions for weak interaction, given in Ref. [46]

$$
\begin{aligned}
e_{\mathrm{LL}}(\gamma) \approx & \gamma - \frac{4}{3\pi}\gamma^{3/2} + \frac{\pi^2 - 6}{6\pi^2}\gamma^2 - \frac{4 - 3\zeta(3)}{8\pi^3}\gamma^{5/2} - \frac{4 - 3\zeta(3)}{24\pi^4}\gamma^3 \\
& - \frac{45\zeta(5) - 60\zeta(3) + 32}{1024\pi^5}\gamma^{7/2} - \frac{3[15\zeta(5) - 4\zeta(3) - 6\zeta^2(3)]}{2048\pi^6}\gamma^4 \\
& - \frac{8505\zeta(7) - 2520\zeta(5) + 4368\zeta(3) - 6048\zeta^2(3) - 1024}{786432\pi^7}\gamma^{9/2} \\
& - \frac{9[273\zeta(7) - 120\zeta(5) + 16\zeta(3) - 120\zeta(3)\zeta(5)]}{131072\pi^8}\gamma^5 + O\left(\gamma^{11/2}\right). \tag{34}
\end{aligned}
$$

It has been shown in [57] that Eq. (34) (with expansion cut after the sixth term [the one with $\gamma^{7/2}$]) taken for $\gamma \ll 1$ matches asymptotically Eq. (33) taken for $\gamma \gg 1$.

## B   Numerical implementation of (LL)GPE

The (LL)GPE (cf. Eq. 7 for GPE and Eq. 15 for LLGPE) is a complex, non-linear partial differential equation. In order to solve it, we use the imaginary time evolution (ITE) method. The function $\phi$ is represented on a one-dimensional spatial lattice with $N_x$ fixed points with lattice constant $\mathrm{DX} = \frac{L}{N_x}$.

The algorithm is built in such way that we choose an initial guess. Generally, it can be any function. We further evolve the initial guess in imaginary time $t \mapsto -i\tau$. It is done with the use of split-step numerical method. The evolution in kinetic energy term is done in the momentum domain, the self-interaction term is calculated in the spatial domain.

No external potential is used. The solution is being found in a box with periodic boundary conditions $\phi\left(\frac{-L}{2}\right) = \phi\left(\frac{L}{2}\right)$.

The program implementing the algorithm above is available here: https://gitlab.com/jakkop/mudge/-/tags/v01Jun2021. The program uses W-DATA format dedicated to store data in numerical experiments with ultracold Bose and Fermi gases. The W-DATA project is a part of the W-SLDA toolkit [58–60].

# C  Phase imprinting method

We use phase imprinting to generate solutions of black solitons; in every iteration $\phi$ is modified in such a way that $\arg \phi(x) = \pi \cdot \left( \frac{x}{L} + 0.5 \right)$ for $x \in \left\langle -\frac{L}{2}, 0 \right\rangle$ and $\arg \phi(x) = \pi \cdot \left( \frac{x}{L} - 0.5 \right)$ for $x \in \left\langle 0, \frac{L}{2} \right\rangle$. One can easily see that there is a $\pi$ phase jump for $x = 0$. Moreover, the periodic boundary conditions in terms of phase, i.e. $\arg \phi \left( \frac{-L}{2} \right) = \arg \phi \left( \frac{L}{2} \right)$, are fulfilled. The soliton moves with constant velocity $v_s = \frac{\hbar}{m} \frac{\pi}{L}$.

It is worth mentioning that the healing length $\xi$ is the proper soliton width scale for $\gamma \ll 1$ (as in the GPE), whereas for $\gamma \gg 10$ (fermionized regime) soliton width scales with average interparticle distance $L/N$.

The resulting soliton profile remains unchanged in the course of the real-time evolution, as can be seen in Figs. 8 and 9.

On the other hand, when we initialize the real time evolution within LLGPE, but with the soliton from GPE, we immediately see it gets distorted. It is shown in Figs. 10 and 11.

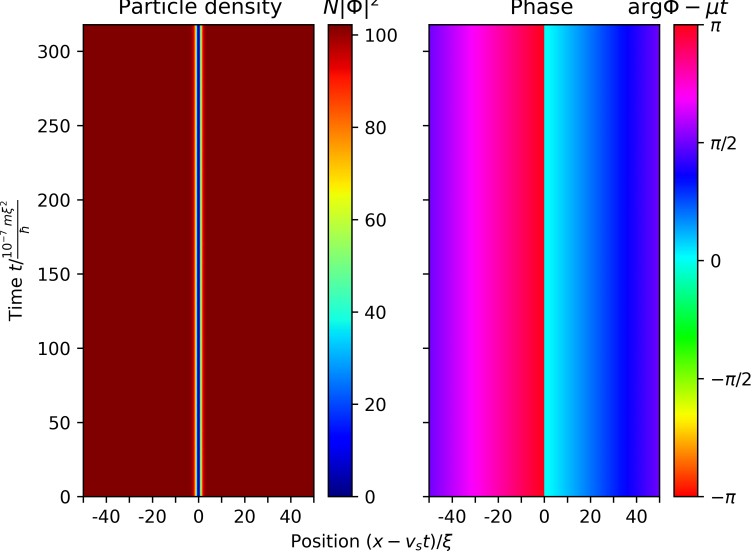

Figure 8: Particle density (left) and phase $\arg \phi$ (right) during the real-time evolution of the LLGPE approach for $\gamma = 1$. Other parameters as in Fig. 4.

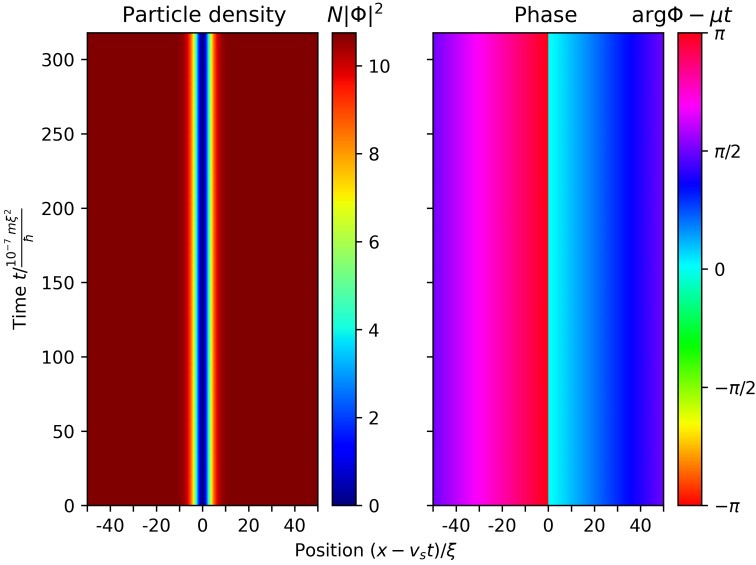

Figure 9: Particle density (left) and phase $\arg\phi$ (right) during the real time evolution of the LLGPE approach for $\gamma = 100$. Other parameters as in Fig. 4.

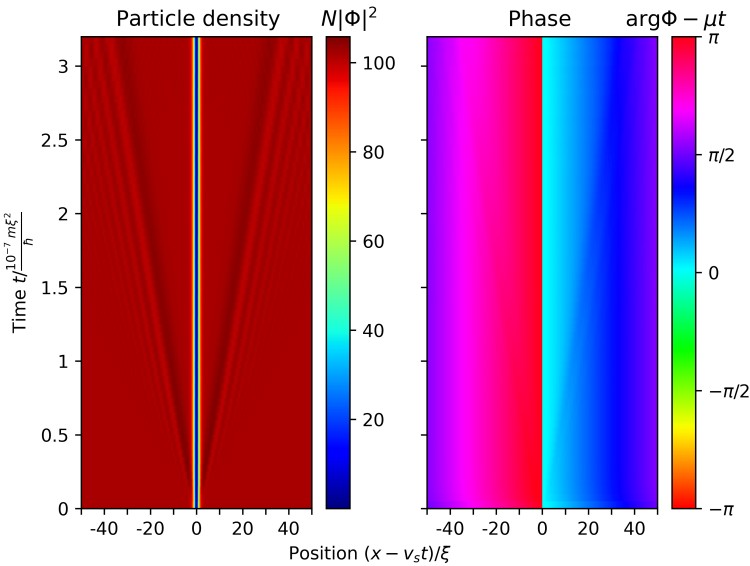

Figure 10: Particle density (left) and phase $\arg\phi$ (right) during the real time evolution of GPE solitonic solution within the LLGPE soliton for $\gamma = 1$. Other parameters as in Fig. 4.

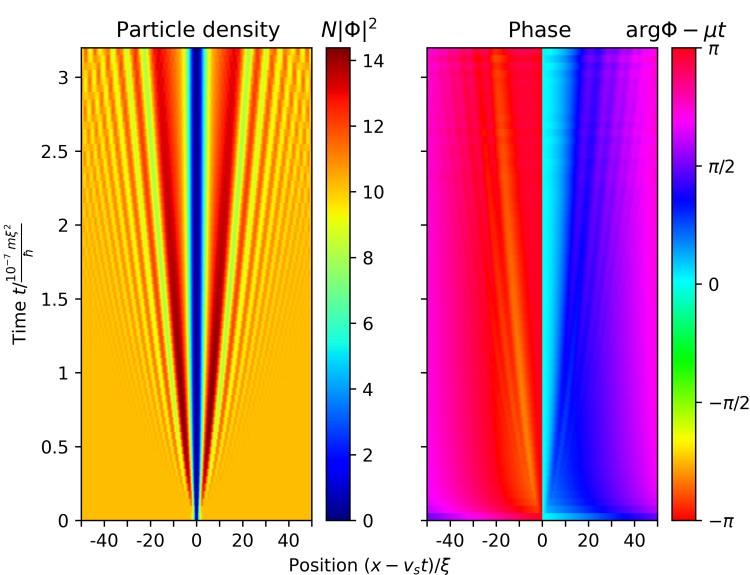

Figure 11: Particle density (left) and phase $\arg \phi$ (right) during the real time evolution of GPE solitonic solution within the LLGPE approach for $\gamma = 100$. Other parameters as in Fig. 4.

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
