# Peer review of "Beyond Gross-Pitaevskii equation for 1D gas: quasiparticles and solitons"

_SciPost Physics, doi:SciPost Phys. 12, 023 (2022)_

## Round 2 · Referee Report · Anonymous (Referee 1) · 2021-8-24

Strengths

see enclosed report

Weaknesses

see enclosed report

Report

See enclosed report

Requested changes

See enclosed report

Attachment

  • validity: high
  • significance: high
  • originality: high
  • clarity: ok
  • formatting: good
  • grammar: good

Author:  Jakub Kopyciński  on 2021-11-03  [id 1907]

(in reply to Report 1 on 2021-08-24)

Dear Referee,

We thank you for careful reading and for the positive opinion. Concerning the comments: 1.

In the abstract the meaning of the so-called type-I and type-II excitations should be briefly clarified. Which are these excitations and what are their differences? Do they refer to particle and hole excitations related to phonons and solitons respectively? Also, an example needs to be provided for the long- and short-wave structures.

We have followed the comment of the Referee and clarified the meaning of the elementary excitations in the abstract of our work. With this respect, we have also changed the graphical abstract and the introduction.

2.

I would strongly encourage the authors to describe in the Introduction in more detail the major results of their study. Currently, the description is very short and it is focused on the topics that will be discussed and not the actual new results. Moreover, since the topic of quantum dark solitons is a main topic of interest in the main text I believe that it deserves to be mentioned even briefly in the Introduction. Here, relevant references besides the ones by Sacha should be included such as earlier studies where some of the authors participated but also from other groups (see e.g. Phys. Rev. A 98, 013632 (2018), New J. Phys. 19, 073004 (2017)). This way, the reader can be directed to them and have an overview both for the homogeneous and for the trapped cases.

The main goal of our work is the analysis of the generalized non-linear equation describing point-like interacting one-dimensional Bose gas for any interaction strength. We address the applicability of the equation and its strong and weak points. The main findings of our work are the following: +Coherent derivation of the LLGPE +Explicit analysis of the particle excitations by linearization of the equation and finding agreement between the LLGPE and the Lieb-Liniger model +Analysis of LLGPE dark solitons with respect to the GPE and the yrast states from the LL model +Analysis of the LLGPE limitations: there are such solutions of the LLGPE that are beyond the equation's validity range. This includes narrow solitons. One thus cannot now judge whether solitons in the strongly interacting 1D gas exist or do not exist. As suggested by the Referee, we have extended the description of the enlisted results in the Introduction. We thank the Referee for their valuable comment. According to their second remark, we have also added the discussion of quantum dark solitons in the first paragraph of the Introduction. From the perspective of the content of our work, the essential part of the body of the exhaustive literature on quantum dark solitons that should be introduced in the manuscript pertains to the correspondence between the effective single-particle Gross-Pitaevski equation for one-dimensional untrapped repulsive gas and the underlying many-body model given by the Lieb-Liniger model. We have added the relevant references providing a better overview of the topic than before.
The references suggested by the Referee study different, important problems concerning solitons with the ML-MCTDHB method. A question arises whether the LLGPE could be also used for investigating an impurity moving in a cold gas or the dynamics of dark-in-bright solitons. Although answering that goes beyond the scope of the current manuscript, this intriguing prospect warrants further study.

3.

In a related note a mentioning on the fact that droplets in one-dimension have been investigated already by using ab-initio approaches such as the quantum monte-carlo is missing. This should be certainly rectified and relevant references in one-dimension such as Phys. Rev. A 102, 023318 (2020), Phys. Rev. Lett. 122, 105302 (2019) and arXiv:2108.00727 need to be included if the authors want to keep the claims regarding the Lee-Huang-Yang energy correction. Currently only references regarding higher dimensions are provided while the focus of the present analysis is in one-dimension.

We thank the Referee for this remark. Indeed, we overlooked citing the relevant papers on 1D systems in the introduction. We have added suitable references. 4.

In the Introduction, the sentence “The figures of merit are their dispersion relations and ...” reads awkwardly. Please rephrase.

We thank the Referee for the meticulous reading of our manuscript. We have corrected the sentence.

5.

What is the physical interpretation of the pressure term given in Equation (4)? Please elaborate.

The role of pressure is analogous to that in classical hydrodynamic equations. To give the reader some intuition about the quantum effects occurring here, we have added after equation (17) a clear interpretation of the origin of this expression in the limit $\gamma\to\infty$, where fermionization occurs. The pressure may be thus seen as fermion degeneracy pressure.

6.

After equation (7) it is stated that the Gross-Pitaevskii theory is not justified for strongly repulsive atoms. I do not entirely agree with this statement. I think it should be complemented and mention that also in the weakly interaction regime the Gross-Pitaevskii theory can fail due to the presence of quantum correlations. There are several demonstrations of this fact especially regarding the dynamics of cold gases.

Indeed, the statement was too strong. We have rephrased the paragraph to stress both the importance of the GPE and its limitation

7.

At the end of equation (10) the “,” should be replaced by “.”

Fixed.

8.

The last terms appearing in equations (13) and (17) and representing beyond Gross-Pitaevskii corrections (if I understand correctly) as well as their role should be clearly explicated. Why these terms operate beyond a macroscopically occupied orbital and for which order/type of correlations do they account for? This is a central finding of the present work and should be clearly communicated to the reader.

This term is taken directly from the exact results of the Lieb-Liniger model. It represents the energy in a small volume in all for any interaction strength. Should we expand it in $\gamma$ around $\gamma\approx 0$, then the zeroth-order would be equivalent to the Gross-Pitaevskii energy. The following term would correspond to LHY correction, and the subsequent terms would mean higher-order corrections. On the other hand for large $\gamma$, one can expand $\mu_{\rm LL}$ in $1/\gamma$, but we do not know how to interpret the expansion terms physically. In particular in the case of strong contact interaction, there is no orbital occupied by almost all atoms, as assumed in the GPE equation. Still, the equation we used to describe the gas density and its velocity can be valid even in this regime.

9.

After equation (17) it is stated that equation (13) coincides with the equation proposed in Ref. [15] in the limit of $\gamma \ll 1$. I suggest to explicitly state some more details here regarding the equation provided in Ref. [15] in order the corresponding description to be accessible to a broad audience.

We agree with the Referee that a short discussion about the limit $\gamma \to \infty$ (we assume a typo in the comment above), including the equation, would be here in place. 10.

At the beginning of Section 4 the word “apperas” should read “appears”.

Fixed

11.

In Section 4.1 the case of a soliton in a Tonks-Girardeau gas is discussed among others. What is the value of the interaction strength g where this strongly coupled bosonic system is approached? It is also very important that the properties of the black solitons in this regime will be mentioned. Do they remain the same as in the weakly interacting case? Please elaborate at least briefly and also provide some relevant references if any to this topic such that the interested reader can be directed.

In section 4.1, we have added information that the Tonks-Girardeau limit is approached for $\gamma\geq 100$. At present, we briefly mention black soliton properties and have added a proper reference.

12.

On page 11, the sentence “We examine closer a relation between these excitations studying ...” is not understandable to me. Please rephrase.

We have removed the sentence and modified the subsequent paragraph accordingly. 13.

The legend of figure 5 is not displayed properly. Please fix this issue.

This problem occurred due to improper figure rendering. It has been already fixed.

14.

In the right panels of figure 5 the two- and three-body correlation functions are shown. It can be seen that in the non-interacting case both the two- and three-body correlations do not become zero at the center of soliton. However, in the Tonks regime there is a clear correlation hole probably due to fermionization. I wonder whether the finite value of $G_2$ and $G_3$ at the center of the soliton in the non-interacting case is related to the fact that quantum dark solitons have a filled core due to presence of correlation effects. Please comment. If this is not the case how the basic properties of quantum black solitons can be discerned within this approach. Here I mean properties such as the filled core of the dark soliton which is a manifestation of the presence of condensate depletion and in turn related for instance to the dispersion of the dark soliton’s position. These processes are already discussed in the already cited references and the ones suggested in question 3. A relevant discussion is necessary.

Indeed, the quantum effects are responsible for the fluctuations of the soliton centre in interacting gas, and, after averaging, the gas density increases inside the solitonic core. This effect was nicely presented in Phys. Rev. A 66, 043620 (2002). However, as shown in Phys. Rev. A 97, 063617 (2018), this effect does not emerge in the limit of vanishing interactions. Therefore, the origin of the effect shown in Fig. 5 is different. We simply presented only the $G_2$ and $G_3$ functions. In the higher-order correlation functions $G_m$ in the limit $m\to\infty$, the density will reach exactly zero (as rigorously proven in Phys. Rev. A 97, 063617 (2018)). 15.

Along the same lines, can the authors comment on the dynamical evolution of usual mean-field black soliton embedded in the LLGPE approach as an initial condition?

A usual mean-field soliton is narrower than the LLGPE soliton. In the dynamics of the system initiated in such a state, the shock waves appear. We have added examples of such dynamics in appendices, for $\gamma = 1$ and $\gamma = 100$.

16.

In the first paragraph of page 13 it is commented that higher order correlation functions have m local minima and not a single one as the common soliton structure. Is this observation a manifestation of the fact that quantum solitons may exhibit a splitting process or a dispersive behavior due to correlations? Please clarify.

**We have shown that the $G_2$ function practically overlaps with the LLGPE soliton. Then $G_m$ will probably look like a collection of $m-1$ such solitons. Our main conclusion from the last plot is that the relationship between LLGPE and the quantum soliton remains a puzzle. The width of the soliton in the strongly interacting gas is of the order of the interparticle distance. Therefore such a state is beyond the validity regime of a hydrodynamical approach. It means that our LLGPE solitons and the solitons presented in Kolomensky et al. may not have a real, physical counterpart in the many-body quantum system. **

17.

I would suggest to carefully check the entire manuscript for typos and grammar errors beyond the ones mentioned above. This is certainly needed and will be definitely beneficial for the reader.

We have reviewed the manuscript.

We thank you once again for reviewing our manuscript and providing the useful recommendations,

Jakub Kopyciński Maciej Łebek Maciej Marciniak Wojciech Górecki Rafał Ołdziejewski Krzysztof Pawłowski

---

## Round 2 · Referee Report · Anonymous (Referee 2) · 2021-9-16

Report

Report on Ms SciPost 2106.15289v2

Beyond Gross-Pitaevskii equation for 1D gas: quasiparticles and solitons

by J. Kopyciński et al.

The authors present a study of linear and non-linear excitations in weekly to strongly interacting one-dimensional Bose gases, comparing exact solutions of the Lieb-Liniger (LL) model with approximate ones obtained through a hydrodynamic approach. They specifically focus on solitary wave solutions of the Gross-Pitaevskii (GPE) and Lieb-Liniger GP (LLGPE) equations in comparison with linear superpositions of type-II excitations of the LL model, between which a close relation has been discussed in numerous previous works, also cited by the authors. The LLGPE, thereby, takes the form of a non-linear Schrödinger equation, where the nonlinearity represents a non-analytic expression of the particle density.

After discussing in detail the respective models and solutions, they present numerical solutions for the dispersion of the type-I excitations and the respective quasiparticles in the GP models. Subsequently, they solve the field models for one-dimensional dark solitons, for different coupling strengths gamma, and compare these with each other, and, in the Tonks-Girardeau (TG) limit, with the solution of Kolomeisky et al. Comparing, furthermore, their excitation energies with those of exact type-II LL eigenstates, they find a qualitatively significantly better correspondence for the LLGPE solitons when going to large gamma. To learn more about the remaining discrepancy at large gamma, they study in more detail the spatial patterns of the respective solitons, specifically, the single-particle densities as well as non-local 2nd- and 3rd-order correlation functions. They can finally show that the width of the LLGPE solitons interpolates between the known GPE and Kolomeisky widths in the limit gamma->0 and ->infinity, respectively.

The work adds new and interesting results concerning the solitonic solutions of the LLGPE. It demonstrates how well the LLGPE solutions capture exactly computable properties of LL solutions, at least those which can be easily derived within a hydrodynamic approach as given by the LLGPE.

The work appears suitable for publication in SciPost Physics as soon as a few points have been clarified in the presentation:

  1. Given the qualitative similarity of the GPE and LLGPE linear dispersions shown in Fig. 2 and the respective discussion at the end of Sect. 3, it appears desirable to also have a plot of the respective sound velocities as functions of gamma as well as of the deviations between the exact and the approximated dispersions (maybe as insets). Is the apparent agreement between LLGPE and LL of the same quality in the low- and high-momentum limits? (The dispersion formula suggests identity at high p and dependence on the speed of sound at low momenta.)

  2. When I came across Fig. 3 the question of how the widths depend on gamma immediately occurred to me. This is, partly, the subject of Fig. 6 and Sect. 5. Maybe pull the direct presentation of Fig. 6 to the front, into 4.2?

  3. In the discussion of the left panel of Fig. 4, the authors point to the agreement between the LLGPE excitation energy and the LL type-II dispersion. They may want to briefly summarize the LL dispersions somewhere, also discussing the momenta where the type-II dispersion goes down again and touches zero at the "umklapp" momentum?

  4. The functions G_2 and G_3 shown in Fig. 5 are not defined properly, i.e., it is unclear for which arguments they have been evaluated, given that they are defined by Eq. (28). In particular, assuming that the G_2 function is evaluated at two different positions, why can I compare it with the LLGPE density in the upper right panel? What would help in this context is to explain in some more detail how I should imagine the soliton solution to come out of the LL eigenstates. The spectral distributions (spectral functions) of the LL excitations calculated by Brand and Cherny show that for large gamma it is hard to distinguish type I and type II excitations, as the full band in between the respective energies is equally filled for a given momentum k while for smaller gamma, still larger than 1, the low-energy solutions are predominantly of type II. Is there any insight on relations between the properties of these excitations and the soliton excitations of the LLGPE?

  5. In the final sentence of Sect. 4, the authors write that it "is alarming in the context of the hydrodynamical origin of the LLGPE" that "the LLGPE soliton is close to [the] G_2 of the fermionized gas". Could this be elaborated on a bit further? Where do I see this proximity/being close? Maybe, the scales in Fig. 5, lower right panel, should be expanded in an inset?

  6. The authors refer to the formation of quantum droplets in a 1D system which have been the subject of intense research recently. For the less specialized reader, it appears desirable to have a slightly more detailed explanation of why the present research supports the theory used for describing quantum droplets. I understand that the NLSE used in Ref. 24 is of the same type as that used here, in the sense that the nonlinearity is at least similar, with a |phi|^6 factor multiplying a non-polynomial function. In addition, however, the droplet theory also includes a non-local dipole-dipole interaction.

First, it should be clarified to what extent the two models are equivalent to each other (e.g., what would happen if one would approximate the e_LL function in the same manner as it was done in Ref. 24). Secondly, it should be said, to what extent probing the solitary wave solutions of the LLGPE in the present ms. supports the specific model used for droplet formation. Are the solitons here of direct relevance for the droplet solutions there? Do they have the same kind of properties? And if yes, which specifically are relevant?

  1. Minor points (typos etc):

  2. footnote 2: it used rough

  3. Sect. 4, 3rd word
  4. p.11, 2nd par, l. 6: hexagons->pentagons
  5. p.11, last sentence.
  6. Fig. 5: legend boxes overlay each other.
  7. Fig. 5, Caption, l.2: dyspersion
  8. Fig. 5, Caption and figure: NI: do the authors mean "weakly interacting"?
  9. Sect. 4, 3rd last line: soliton.Actually
  10. Sect. 5, l.6: larger that
  11. Sect. 5, 1st par, 2nd last line: around gamma=1/2: the authors mean "gamma ~ 1", I presume (the crossing is at gamma~pi).
  12. Sect. 6, l.3: fundaments->foundations?
  • validity: top
  • significance: top
  • originality: high
  • clarity: high
  • formatting: excellent
  • grammar: good

Author:  Jakub Kopyciński  on 2021-11-03  [id 1908]

(in reply to Report 2 on 2021-09-16)

Dear Referee,

We thank you for careful reading and for the positive opinion. Concerning the comments:

  1. Given the qualitative similarity of the GPE and LLGPE linear dispersions shown in Fig. 2 and the respective discussion at the end of Sect. 3, it appears desirable to also have a plot of the respective sound velocities as functions of gamma as well as of the deviations between the exact and the approximated dispersions (maybe as insets). Is the apparent agreement between LLGPE and LL of the same quality in the low- and high-momentum limits? (The dispersion formula suggests identity at high p and dependence on the speed of sound at low momenta.

We fully agree with this comment. We have added a next figure (Fig.~3 in the current version), with two panels. In the left panel, we show the speeds of sound: the exact result, equivalent to LLGPE and an outcome of linearization of the GPE. In the right panel, we show relative deviations between excitation energy obtained from the LLGPE versus the exact result of the LL model. Indeed, the relative error for high momenta and low is tiny. Significant deviations are present for the intermediate momenta.

2.

When I came across Fig. 3 the question of how the widths depend on gamma immediately occurred to me. This is, partly, the subject of Fig. 6 and Sect. 5. Maybe pull the direct presentation of Fig. 6 to the front, into 4.2?

We have followed this advice. Indeed, we have discussed the widths already in Sec. 4.2. 3.

In the discussion of the left panel of Fig. 4, the authors point to the agreement between the LLGPE excitation energy and the LL type-II dispersion. They may want to briefly summarize the LL dispersions somewhere, also discussing the momenta where the type-II dispersion goes down again and touches zero at the "umklapp" momentum?

To address the comment (of both referees), we decided to give more details about the type-II excitations already in the introduction. To this aim, we improved the graphical abstract and described it in the text. We use more familiar names of these excitations (hole and particle excitations) already in the abstract. 4.

The functions $G_2$ and $G_3$ shown in Fig. 5 are not defined properly, i.e., it is unclear for which arguments they have been evaluated, given that they are defined by Eq. (28). In particular, assuming that the $G_2$ function is evaluated at two different positions, why can I compare it with the LLGPE density in the upper right panel? What would help in this context is to explain in some more detail how I should imagine the soliton solution to come out of the LL eigenstates. The spectral distributions (spectral functions) of the LL excitations calculated by Brand and Cherny show that for large gamma it is hard to distinguish type I and type II excitations, as the full band in between the respective energies is equally filled for a given momentum k while for smaller gamma, still larger than 1, the low-energy solutions are predominantly of type II. Is there any insight on relations between the properties of these excitations and the soliton excitations of the LLGPE?

We thank the Referee for these comments. Firstly, we clarified relations between the GPE /LLGPE solitons and the type-II excitations, adding two paragraphs of explanations to the text. The solitons should appear out of the type-II (solitonic) excitations in the course of the particle measurement. The measurement is understood in a broader sense, and it may be performed spontaneously, for instance, by particle losses (as discussed in Phys. Rev. Research 2, 033368 (2020)). The other typical way is just absorption imaging, which measures a high-order correlation function. On the other hand, the low weight of solitons in the spectral function for large $\gamma$ indicate that other methods, like the Bragg scattering, may be insensitive to solitons. 5.

In the final sentence of Sect. 4, the authors write that it "is alarming in the context of the hydrodynamical origin of the LLGPE" that "the LLGPE soliton is close to [the] $G_2$ of the fermionized gas". Could this be elaborated on a bit further? Where do I see this proximity/being close? Maybe, the scales in Fig. 5, lower right panel, should be expanded in an inset?

In the very last stage of the manuscript preparation, we introduced a bug in the figure. We are very sorry for this. Now, the figure is corrected. We hope that it is clear now that $G_2$ coincides with the LLGPE soliton density. Then $G_3$, with two density dips is of course totaly different. 6.

The authors refer to the formation of quantum droplets in a 1D system which have been the subject of intense research recently. For the less specialized reader, it appears desirable to have a slightly more detailed explanation of why the present research supports the theory used for describing quantum droplets. I understand that the NLSE used in Ref. 24 is of the same type as that used here, in the sense that the nonlinearity is at least similar, with a $|\phi|^6$ factor multiplying a non-polynomial function. In addition, however, the droplet theory also includes a non-local dipole-dipole interaction.

Indeed, this remark was solely related to Ref. [24]. Droplets found in [24] arise due to the competition between the local and dipolar forces AND the contribution of the beyond mean-field effects to the energy coming from strong local interactions. The role of the latter corrections is crucial for droplet formation. In this manuscript, we wanted to test such beyond mean-field corrections (also beyond Bogoliubov). Moreover,  in [24], we proposed the equation referring to the local density approximation. This manuscript's purpose was to understand the equation better, find appropriate interpretations, and benchmark it with an exact solution. We changed the Conclusions to indicate precisely the LLGPE scope and its relation to the work about the mentioned quantum droplet.

First, it should be clarified to what extent the two models are equivalent to each other (e.g., what would happen if one would approximate the $e_LL$ function in the same manner as it was done in Ref. 24). Secondly, it should be said, to what extent probing the solitary wave solutions of the LLGPE in the present ms. supports the specific model used for droplet formation. Are the solitons here of direct relevance for the droplet solutions there? Do they have the same kind of properties? And if yes, which specifically are relevant?

We had done such benchmarks way before submitting this manuscript. The model used in [24] is qualitatively but not quantitatively correct. We found the comparison, however, not appropriate for the core part of this manuscript. In the present version, we have commented on the relations and added comparisons to an appendix. The only role of solitons in this respect is in the discussion of the validity of the LLGPE. The hydrodynamical equations underlying the LLGPE proved useful for long-wave structures like phonons and limited applicability for short-wave ones like narrow solitons. Concerning the extension of the LLGPE with the dipolar interaction -- the droplet solution and its low-energy excitations should be in the validity range of the hydrodynamical approach. Still, we are trying to use more sophisticated methods, like DMRG, to check numerically the validity of solutions of the LLGPE extended by non-local interactions. That study is, however, beyond the scope of this manuscript.

7.

Minor points (typos etc): - footnote 2: it used rough - Sect. 4, 3rd word - p.11, 2nd par, l. 6: hexagons->pentagons - p.11, last sentence. - Fig. 5: legend boxes overlay each other. - Fig. 5, Caption, l.2: dyspersion - Fig. 5, Caption and figure: NI: do the authors mean "weakly interacting"? - Sect. 4, 3rd last line: soliton.Actually - Sect. 5, l.6: larger that - Sect. 5, 1st par, 2nd last line: around gamma=1/2: the authors mean "gamma ~ 1", I presume (the crossing is at gamma~pi). - Sect. 6, l.3: fundaments->foundations?

We thank you for your very careful reading of our manuscript and are sorry for so many misprints. Concerning: "Fig. 5, Caption and figure: NI: do the authors mean "weakly interacting" The result is practically the same for very weakly interacting gas (with the healing length larger than the size of the box $L$) and for the ideal gas. Although there are no solitons in the limit $g\to0$, soliton-like states exist. We discuss this paradox in Phys. Rev. A 97, 063617 (2018). We used results for the ideal gas, knowing that they do not differ from the weakly interacting system. We have corrected the rest of the typos.

We thank you once again for reviewing our manuscript and providing the useful recommendations,

Jakub Kopyciński Maciej Łebek Maciej Marciniak Wojciech Górecki Rafał Ołdziejewski Krzysztof Pawłowski

---

## Round 3 · Referee Report · Anonymous (Referee 2) · 2021-11-7

Report

The authors have carefully answered all my comments and amended their manuscript correspondingly.

I am happy to recommend publication in SciPost Physics once the authors have considered a few remaining/newly arising questions:

  1. Once again to the formulation that "..the LLGPE soliton is close to G_2 of the fermionized gas is alarming in the context of the hydrodynamical origin.." at the end of Sect. 4.

I now see the proximity of the G_2 and the soliton solution, what I do not understand, yet, though, is why the authors consider this an "alarming" observation. Do they mean "interesting" or "striking"? Otherwise, if one should worry about this proximity, it would be useful to have an explanation. I would expect that in the Tonks limit, a soliton should not have a spatial profile larger than the interaction radius of the atoms. As the fermionization tells, particles no longer want to come close to each other, so that explains the shape of the G_m for any m, as the authors point out. But shouldn't a soliton in the Tonks limit be something like a fermionic hole excitations, possibly arising out of a pair formation process? In that respect I would not consider this finding as alarming but as consistent with the expectation, or, thinking closer about it, as interesting, given the fact that type I and type II branch particles at a particular momentum are not really separate excitations in the TG limit (taking into account the findings of Brand and Cherny again)? Maybe the authors could provide still a few clarifying remarks?

  1. A technical question in this context: Where is x' in the top right panel of Fig. 7? It is indicated in the TG limit in the lower right panel, but not in the NI limit. Do I have to consider, here, x' to be given by the shown distribution of G_2?

Further to the NI limit: I understand that there should be soliton-like solutions in the ideal gas, if there are such in the WI limit of any small gamma. But shouldn't such solutions correspond to product states of single-particle wave functions showing a dip at a particular point. In their cited paper PRA97, 06317 (2018), the authors derive such shapes which just look like that: cosine waves in the periodic boundary conditions which have a dip in the center of the volume. I would imagine a product of such free wave functions to form something like a dark soliton. And as I take from that paper, these "solitons" are in fact dispersive. So should one really talk of "solitons" in that limit? I find this a bit misleading and at least requiring explanations.

  1. A few more minor points (typos etc):

  2. Intro, 1st sentence: In weakly....Bose gases, ...

  3. p. 4, 4th last line: "find to be useful" -> "are found to be useful"?
  4. p. 7, par. before Eq. 18: occurs in ... Bose gases
  5. p. 12, 3rd last line: "in the log.." -> "on a double-log..."
  6. p. 13, 2nd last line of 1st par, Sect. 4.2: static: stationary?
  7. p. 14, 1st line: to infinity
  8. p. 14, 4th last par., 2nd line: authors
  9. p. 16, Sect. 6, 4th l.: giving stronger foundations for further extensions, like...?
  10. dois missing in Refs. 9,10,17-20,25
  • validity: -
  • significance: -
  • originality: -
  • clarity: -
  • formatting: -
  • grammar: -

Author:  Jakub Kopyciński  on 2021-11-29  [id 1991]

(in reply to Report 1 on 2021-11-07)

Dear Referee,

let us provide you with our reply to your questions and comments.

1.

Once again to the formulation that the LLGPE soliton is close to $G_2$ of the fermionized gas is alarming in the context of the hydrodynamical origin at the end of Sect. 4. I now see the proximity of the $G_2$ and the soliton solution, what I do not understand, yet, though, is why the authors consider this an "alarming" observation. Do they mean "interesting" or "striking"? Otherwise, if one should worry about this proximity, it would be useful to have an explanation. I would expect that in the Tonks limit, a soliton should not have a spatial profile larger than the interaction radius of the atoms. As the fermionization tells, particles no longer want to come close to each other, so that explains the shape of the $ G_m $ for any $m$, as the authors point out. But shouldn't a soliton in the Tonks limit be something like a fermionic hole excitations, possibly arising out of a pair formation process? In that respect I would not consider this finding as alarming but as consistent with the expectation, or, thinking closer about it, as interesting, given the fact that type I and type II branch particles at a particular momentum are not really separate excitations in the TG limit (taking into account the findings of Brand and Cherny again)? Maybe the authors could provide still a few clarifying remarks?

Finally, we change "alarming" to "striking".

Our problem with this overlap between solitons and $G_2$ function in the TG regime is twofold: a) the relation between the yrast state and the soliton is completely different that to what we used to in the weakly interacting regime. For weak interaction, the yrast state is close to a continuous superposition of mean-field solitons, with a random position. It doesn't seem the case in the TG regime. b) As written in the text, this coincidence happens beyond the hydrodynamic equation, namely, it is impossible to define "fluid elements" consisting of many atoms building the solitonic profile.

But indeed, in the LL quasimomenta picture, a soliton is like particle-hole excitation. Also, in the TG regime, one can easily construct type I excitations from type II ones.

2.

A technical question in this context: Where is $x'$ in the top right panel of Fig. 7? It is indicated in the TG limit in the lower right panel, but not in the NI limit. Do I have to consider, here, $x'$ to be given by the shown distribution of $G_2$?

We replot the figure to make everything clear. We indicate $x'$ in both panels. Indeed $x'$ needs to be given by the shown distribution of $G_2$.

Further to the NI limit: I understand that there should be soliton-like solutions in the ideal gas, if there are such in the WI limit of any small gamma. But shouldn't such solutions correspond to product states of single-particle wave functions showing a dip at a particular point. In their cited paper PRA97, 06317 (2018), the authors derive such shapes which just look like that: cosine waves in the periodic boundary conditions which have a dip in the center of the volume. I would imagine a product of such free wave functions to form something like a dark soliton. And as I take from that paper, these "solitons" are in fact dispersive. So should one really talk of "solitons" in that limit? I find this a bit misleading and at least requiring explanations.

Strictly speaking the type II excitations in the NI limit are highly entangled (due to symmetrization) states. Hence they cannot be reduced to simple product states. But once the symmetry is broken, the system may fall into a product state, in which all particles occupy a density dip. After this symmetry breaking procedure, the dip moves with a constant velocity, preserving its shape.

In our paper we have also shown that the yrast states corresponding to gray solitons can be considered as composed of mean-field solitons, but with different momenta. In each symmetry breaking event one find a mean-field soliton, but with a random depth. 3.

A few more minor points (typos etc): - Intro, 1st sentence: In weakly....Bose gases, ... - p. 4, 4th last line: "find to be useful" -> "are found to be useful"? - p. 7, par. before Eq. 18: occurs in ... Bose gases - p. 12, 3rd last line: "in the log.." -> "on a double-log..." - p. 13, 2nd last line of 1st par, Sect. 4.2: static: stationary? - p. 14, 1st line: to infinity - p. 14, 4th last par., 2nd line: authors - p. 16, Sect. 6, 4th l.: giving stronger foundations for further extensions, like...? - dois missing in Refs. 9,10,17-20,25

These minor changes will be also included before the publication.

Thank you once again for the careful reading of our manuscript, Jakub Kopyciński Maciej Łebek Maciej Marciniak Wojciech Górecki Rafał Ołdziejewski Krzysztof Pawłowski

---

## Round 3 · Referee Report · Anonymous (Referee 1) · 2021-11-19

**Second referee report SciPost Physics on: ''Beyond Gross-Pitaevskii equation for 1D gas: quasiparticles and solitons" by Jakub Kopycinski, Maciej Lebek, Maciej Marciniak, Rafal Oldziejewski, Wojciech Gorecki, Krzysztof Pawlowski.**

As I mentioned in my original report I find the present work an interesting attempt towards setting up beyond mean-field models so as to deepen our understanding of many-body effects. Additionally, I find that the resubmitted manuscript has been improved since the authors have tried to the best of their ability to address most of the comments raised. Particularly, most of the essential points of my original report have been adequately addressed both within the manuscript as well as in the referee report. For instance, further explanations on limiting cases along with their physical interpretation, such as the pressure term, are now provided within the text. Moreover, more references have been included in the updated version and several typos have been corrected.

I find that the revised manuscript is now elevated and can be accessed by a broader audience. Due to the aforementioned reasons, I can now recommend the revised manuscript as suitable for publication in SciPost Physics.

---

## Round 3 · List of Changes

1. Abstract improvement.
  2. Extension of the reference list.
  3. Clarification of the text in Sec. 4.2.
  4. Addition of Figs. 3, 8-11.
  5. Correction of minor typos and issues.

---

## Editorial Decision

published